# Incrimination of shrews as a reservoir for Powassan virus

Heidi K. Goethert [1✉], Thomas N. Mather[2], Richard W. Johnson[3] & Sam R. Telford III[1]

Powassan virus lineage 2 (deer tick virus) is an emergent threat to American public health, causing severe neurologic disease. Its life cycle in nature remains poorly understood. We use a host-specific retrotransposon-targeted real time PCR assay to test the hypothesis that white-footed mice, considered the main eastern U.S. reservoir of the coinfecting agent of Lyme disease, is the reservoir for deer tick virus. Of 20 virus-infected host-seeking nymphal black-legged ticks 65% fed on shrews and none on mice. The proportion of ticks feeding on shrews at a site is positively associated with prevalence of viral infection, but not the Lyme disease agent. Viral RNA is detected in the brain of one shrew. We conclude that shrews are a likely reservoir host for deer tick virus and that host bloodmeal analysis can provide direct evidence to incriminate reservoir hosts, thereby promoting our understanding of the ecology of tick-borne infections.

[1] Tufts University, Cummings School of Veterinary Medicine, Grafton, MA, USA. [2] University of Rhode Island, Kingston, RI, USA. [3] Martha's Vineyard Tick-borne Illness Reduction Initiative, West Tisbury, MA, USA. ✉email: Heidi.Goethert@tufts.edu

Powassan virus is a member of the tick-borne encephalitis virus complex (Flaviviridae), many of which cause a severe meningoencephalitis. Powassan encephalitis is a devastating disease with a 15% case fatality rate; survivors generally have serious long term neurologic sequelae. There are two distinct Powassan virus lineages: lineage 1 (referred to as prototype Powassan virus, POWV), which is found in North America and far Eastern Asia, and lineage 2 (referred to as deer tick virus, DTV) which has only been found in North America[1]. Before 2009[2], all tick-borne encephalitis cases in North America have been attributed to POWV, because etiology has usually been described by the presence of specific antibody and the two viruses are serologically indistinguishable[3]. DTV was demonstrated as a human pathogen when its nucleic acid sequences were amplified from the brain of a fatality (3) and subsequently from other human cases[2,4]. More than 106 cases of POWV encephalitis were reported during 2003–2018, by contrast with 27 cases diagnosed from 1958–1998[5,6]. The recent emergence of Powassan encephalitis[2,5] is likely due to the transmission of the DTV lineage by the aggressively human-biting vector of the agent of Lyme disease, the black-legged tick (*Ixodes scapularis*)[7]. Although this apparent increase in incidence could be attributable to enhanced surveillance for arboviral encephalitides, DTV zoonotic risk is likely increasing as has that for Lyme disease and the other tick-transmitted infections[8]. However, our capacity to predict the extent of its emergence is hindered by an incomplete understanding of the DTV enzootic cycle.

Early studies of the ecology of POWV identified the groundhog tick (*Ixodes cookei*) and squirrel tick (*I. marxi*) naturally maintaining the virus with woodchuck, other medium-sized mammals such as skunks and raccoons, or squirrels serving as likely reservoir hosts, defined herein to be amplifying hosts for the virus[9]. By contrast, DTV has been thought to have a distinct enzootic cycle because it has only been detected in black-legged ticks. Because this tick is the vector for the agents of Lyme disease (*Borrelia burgdorferi*), human babesiosis (*Babesia microti*), and granulocytic anaplasmosis (*Anaplasma phagocytophilum*), and they all share the white-footed mouse (*Peromyscus leucopus*, hereafter referred to as mouse) as a primary reservoir host, the DTV reservoir has been assumed to be identical[10]. There is, however, no direct evidence of their reservoir capacity: virus has never been detected in wild mice, nor has infection been demonstrated in ticks that had fed on mice. We report incrimination of a likely vertebrate reservoir of DTV by analyzing host-seeking infected nymphal ticks for evidence of the identity of the host providing the infectious bloodmeal in the preceding larval stage.

## Results and discussion
Samples of host-seeking nymphal black-legged ticks were collected during 2018–2020 from Massachusetts and Rhode Island sites where DTV is enzootic (Fig. 1). Individual DTV-infected ticks were identified by RT-PCR; all ticks (including virus-negative ticks) were also analyzed for *B. burgdorferi* infection by PCR. Host bloodmeal remnant identification using assays targeting family and order specific retrotransposons was performed as described[11], with the addition of newly described primers (see Table 1). Assays targeted likely mammalian reservoir hosts within our study sites.

We identified 20 nymphal ticks that contained DTV RNA from 13 different sites (prevalence, 0.4–7%, Table 2) and confirmed viral identity by sequencing a 248 bp section of the *NS5* gene and 286 bp section of the *envelope* gene. Cognate viral sequences from these ticks were assigned to the DTV lineage (Fig. 2). Sequences from ticks collected from field sites in close geographic proximity

often clustered together. *Borrelia burgdorferi* prevalence was more variable, ranging from 0 to 21%., and *B. burgdorferi* infection was not associated with DTV infection ($p = 0.5$, Fig. 3), as sites with high numbers of ticks infected with spirochetes were not the same as those that had high numbers of DTV-infected ticks (Table 2).

The source of the infectious larval bloodmeal was identified from 16 of the 20 DTV-infected ticks (80%), 13 of which were identified as shrews (65%) (Table 3). The other DTV-infected ticks had fed on diverse hosts such as bird, squirrel, and cat. One tick showed evidence of having fed on multiple hosts (shrew and deer). None of the ticks had fed on a mouse. We conclude that in our sites, during the years that we sampled, larval ticks feeding on shrews were more likely to be infected by DTV than by feeding on any other animal. Using the 0.1% estimated rate of transmission of adult female ticks to larval progeny for the related tick-borne encephalitis virus[12], we calculated that up to four ticks (95% binomial confidence interval of 0 to 0.4% of ticks) from our study could derive from inheritance. Thus, we cannot exclude this as the source of the single infected ticks derived from a bird, squirrel, and cat. However, more than four infected ticks were derived from shrews, suggesting that inheritance alone cannot explain the apparent association.

During the years that we sampled our study sites, mice did not contribute as many larval bloodmeals as might be expected[13,14]. The proportion of nymphal ticks that fed on mice ranged from 2 to 20% (median 10.5%) (Table 4). Our previous publication identified sites where the majority of ticks had fed on mice (Nantucket 2018, 100%, and Robin's Island 2018 and 2019, 91% and 53%, respectively[11]), but DTV was not identified from these collections. Squirrels, or other *Sciuridae*, contributed ticks at only two sites (median host contribution, 1%). In contrast, shrews were common hosts at our study sites, with a median host contribution of 40.5% (range, 0–68%). The proportion of nymphal ticks that fed on shrews as larvae at a site was associated with the prevalence of DTV infection in ticks at that site ($R^2 = 0.44$ $p = 0.01$, Fig. 3b), but not the prevalence of *B. burgdorferi* ($R^2 = 0.04$, $p = 0.5$, Fig. 3a). DTV-infected nymphs were highly likely to had fed on a shrew (OR = 139, 95% confidence interval 42–456, but not a mouse, squirrel (or other *Sciuridae*) or other host (Fig. 4a). By contrast, *B. burgdorferi*-infected ticks were likely to have fed on mice, but not shrews (OR = 1.1, 95% confidence interval 0.6–1.9) (Fig. 4b). This excludes the hypothesis that shrews were found to have served as virus sources simply because these hosts were the dominant host in these sites.

Three *B. brevicauda* shrews were trapped from two of our study sites in September of 2020. DTV was detected in the brain of one shrew. Attempts to isolate virus by suckling mouse inoculation failed. Sequencing of two gene targets demonstrates greatest similarity to virus found in a tick from the same site (Fig. 2) and not to standard laboratory strains.

DTV, like other tick-borne encephalitis viruses, may be perpetuated by three mechanisms[15]. Virus may be inherited by the tick, transovarial transmission[16]. We found that a greater number of ticks were associated with a specific host from all study sites than expected by vertical transmission, indicating that these ticks were not likely to have inherited the infection. There may be co-feeding or nonsystemic transmission in which an infected tick may serve as the direct source of infection for uninfected ticks attached to the skin around it, with no requirement for hematogenous viral dissemination[17]. Finally, horizontal transmission, in which a larval tick acquires infection from a viremic vertebrate host, requires a reservoir host that is susceptible to infection and allows for sufficient viremia to infect ticks as well as being sufficiently infested by the tick vector[16]. We focused solely on host-seeking nymphal ticks because they would only have one bloodmeal source, that of the larvae. Although adult ticks are also

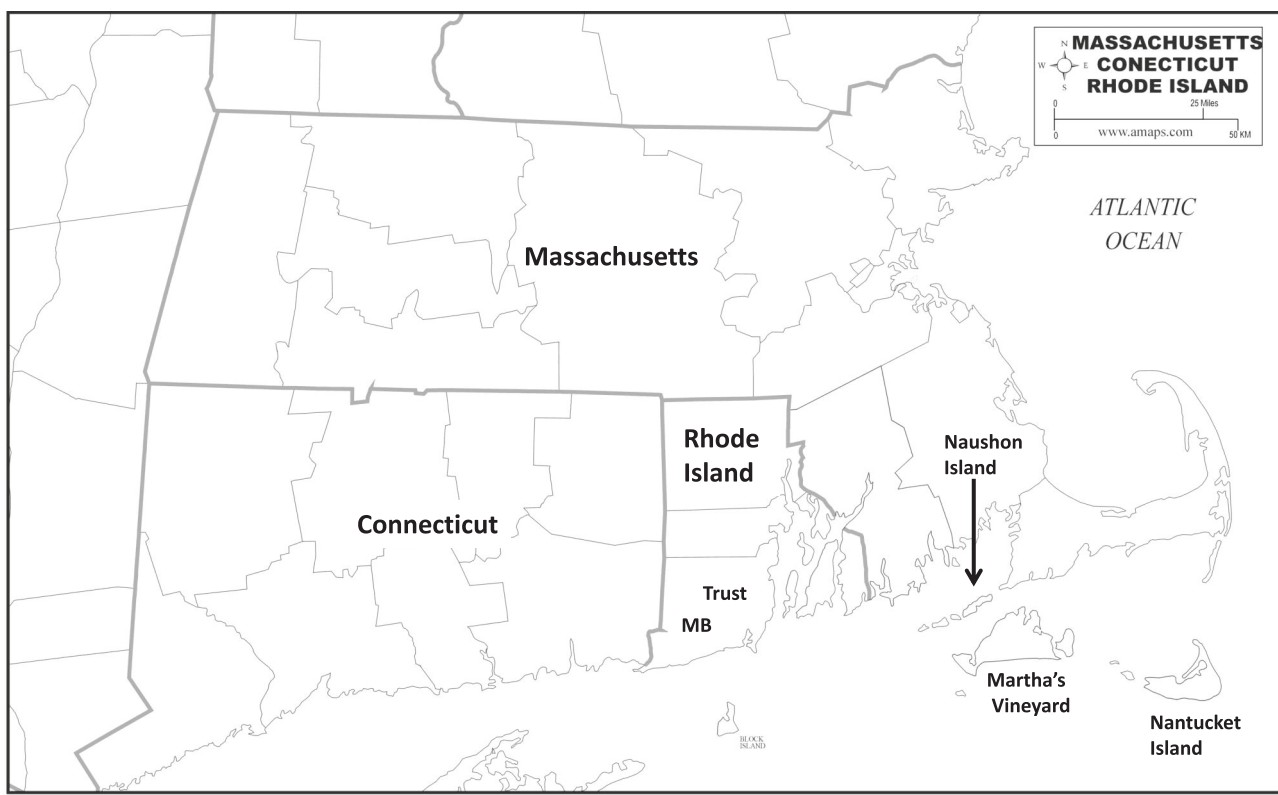

**Fig. 1 Map of the field sites included in this study.** Ticks were collected from two sites in Washington County, Rhode Island, MB and Trust, as well as, from three islands off the coast of Massachusetts: Nantucket, Martha's Vineyard, and Naushon Island.

| Host targeted | SINE family | Primer name | Fluorophore attached to probe | Sequence of primers and probes | Sensitivity[c] | Host specificity[d] |
|---|---|---|---|---|---|---|
| **Table 1 Primers and probes targeting mammalian retrotransposons used in the study for bloodmeal identification in ticks.** | | | | | | |
| *Multiplex 1* | | | | | | |
| Mouse[a] | B1 | PISINE-41F | | GATCTCTGTGAGTTCGAGG | 10⁻⁹ ng/µl | *Peromyscus* |
| | | PISINE-114R | | GTTTCTCTGTGTAGCTTTGC | | |
| | | PISINE-66P | FAM | TGGGCTACCAAGTGAGCTCCAGG | | |
| Rabbit | C48 | RabbSINE-131F-AG | | GGAAGGCAGTGGAGGAT | 10⁻⁹ ng/µl | *Lagomorpha* |
| | | RabbSINE-196R-TGG | | GGTGCTTCCTCCTGGTCT | | |
| | | RabbSINE-167P | Cy5 | GGGCCCTGCACCCCATGG | | |
| Vole | B2 | MicrB2SINE-55F | | TGAGTTCAATTCCCAGCAAC | 10⁻⁷ ng/µl | *Arvoclinae*[b] |
| | | MicrB2SINE-160R | | TGTATACAATATTCTGTCTGTGTG | | |
| | | MicroB2SINE-113P | HEX | GCCCTCTTCTGGCCTGCAGA | | |
| *Multiplex 2* | | | | | | |
| Shrew | SA_Ba-325L16 | SorexSINE-164F | | GATTCCCAGCATCCCATATG | 10⁻⁸ ng/µl | *Soricidae* |
| | | SorexSINE-235rmod | | RTTACTCCTGGCTCTGCA | | |
| | | SorexSINE-185PCA | HEX | GTCCCCCAAGCACCGCCAGG | | |
| Squirrel | CM55 | SqrlSINE-103F | | CCCTGTCTCT AAATAAAATA CA | 10⁻⁷ ng/µl | *Sciuridae* |
| | | SqrlSINE-183R | | TACCAGGGATTGAACTCAG | | |
| | | SqrlSINE-132P | Cy5 | GGGCTAGGGATGTGGCTCAGTGG | | |
| *Single reactions* | | | | | | |
| Deer[a] | Unknown | WTDrep10-167F | Evagreen | GATCTGTTTCACCCTAGATAAT | 10⁻⁸ ng/µl | *Odocoileus* |
| | | WTDrep10-220R | | ATGTTTCAAGAGAACAGCATC | | |
| Cat | SINE | CatSINE-120F | | CTTCGGATTCTGTGTCTCC | 10⁻⁷ ng/µl | *Felis* |
| | | CatSINE-192R | | TATTTTTGGGACAGAGAGAGAC | | |
| | | CatSINE-146P MGB | FAM | TCTGMCCCTCCCCCGTT-MGB | | |
| Bird | CR1-AVI | CR1AVI-23F | | SAGGCCCTGGCACAGG | See Table 5 | |
| | | CR1AVI- 78 R | | CCTTGRACACTTCCAGGGAT | | |
| | | CR1AVI- 35 P | FAM | GCCCAGAGMAGCTGTGGC | | |

[a]previously published (*11*).
[b]These primers cross-react with *Peromyscus* at a concentration of 10⁻⁴ ng/µl.
[c]Sensitivity of each primer set was determined using serial dilutions of positive control DNA. Listed is the last dilution that tested positive.
[d]Host specificity for each primer set was determined by testing primers against a panel of positive control DNA from likely hosts that are common in New England forests at a dilution of 10⁻⁴ ng/µl.

**Table 2 Infection rate of deer tick virus (DTV) and *Borrelia burgdorferi* in ticks at each study site.**

| Site | Year | No. tested | DTV | B. burgdorferi |
|---|---|---|---|---|
|  |  |  | No. pos (%) | No. pos (%) |
| Naushon | 2018 | 28 | 1 (4) | 1 (4) |
|  | 2020 | 28 | 2 (7) | 3 (11) |
| Martha's Vineyard |  |  |  |  |
| VH | 2019 | 57 | 3 (5) | 3 (5) |
| Chil-1 | 2019 | 47 | 1 (2) | 0 |
| Chil-2 | 2020 | 38 | 1 (3) | 7 (18) |
| Chil-3 | 2020 | 29 | 1 (3) | 6 (21) |
| Chap-1 | 2019 | 3 | 1 (nd[a]) | 0 |
| Chap-2 | 2020 | 69 | 4 (6) | 7 (10) |
| WT | 2019 | 54 | 1 (2) | 11 (20) |
|  | 2020 | 76 | 1 (1) | 13 (17) |
| AQ | 2020 | 19 | 1 (5) | 4 (21) |
| Nantucket | 2020 | 108 | 1 (1) | 13 (12) |
| Rhode Island |  |  |  |  |
| MB | 2020 | 197 | 1 (0.5) | 20 (10) |
| Trust | 2020 | 237 | 1 (0.4) | 25 (11) |

[a]*nd* not done.

infected by DTV, they would have had two opportunities to become infected (a bloodmeal during the larval as well as the nymphal stage) and it would not be possible to determine whether the bloodmeal host that was identified from an adult was the source of the virus. Accordingly, we did not analyze adult ticks. Our analysis thus incriminates horizontal transmission between shrews and larval ticks, but we cannot exclude co-feeding transmission.

Shrews (likely *Blarina brevicauda*, the most common shrew in our study sites; our retrotransposon assay, however, may also detect *Sorex* spp.) were the larval bloodmeal host for the majority (65%) of DTV-infected ticks. The infected ticks were collected from eight different sites over the course of three field seasons, indicating that the finding is not spatiotemporally specific. Although our sample size is small, the positive association between the proportion of shrew-fed ticks and the prevalence of DTV infection in ticks also supports a general finding; no association was found between DTV-infected ticks and either mouse-fed or Sciuridae-fed ticks. Finally, we detected virus in the brain of a shrew and find that it is genetically similar to virus within ticks from that site. Shrews are thus the main candidate for the vertebrate DTV reservoir but we cannot now rank the contribution of horizontal transmission relative to other modes of perpetuation. Shrews may be more likely to sustain an infectious viremia,

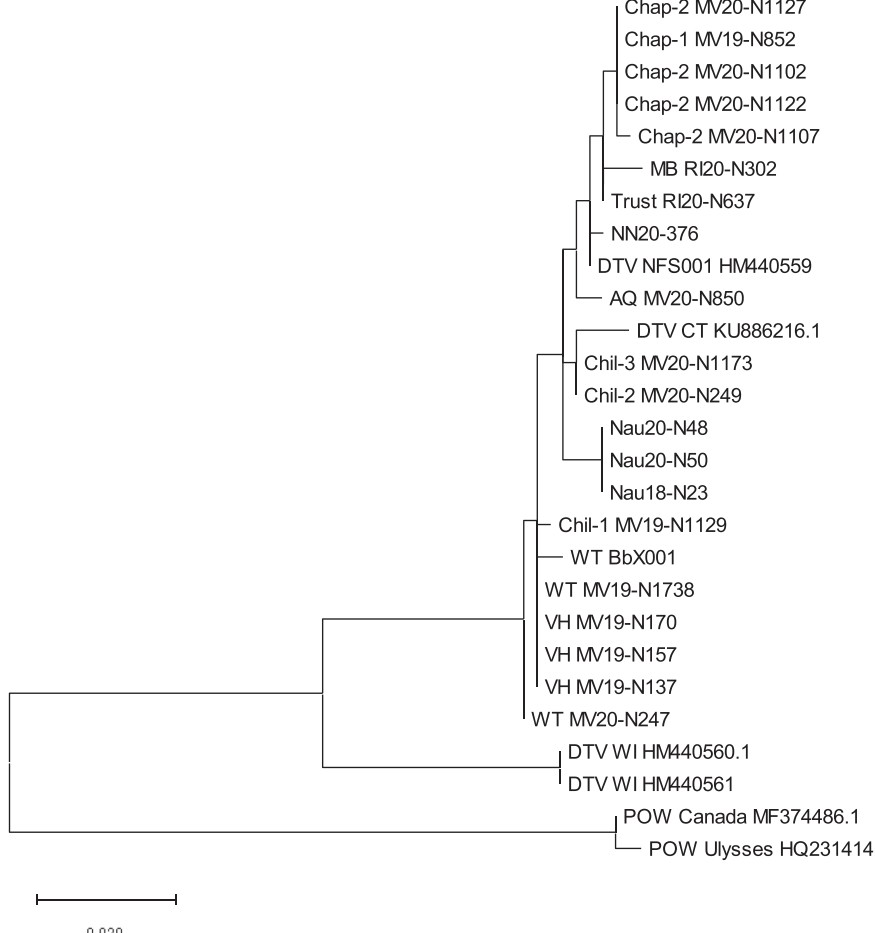

**Fig. 2 Maximum likelihood tree of deer tick virus (DTV) detected in this study.** A 248 bp piece of the *NS5* gene and the 286 bp piece of the *envelope* gene were sequenced from each positive tick in the study, as well as the positive shrew. These pieces were concatenated and aligned with deer tick virus (DTV) and Powassan virus (POW) sequences downloaded from GenBank (GenBank numbers are listed on the tree). A maximum likelihood tree was then created using MEGAX.

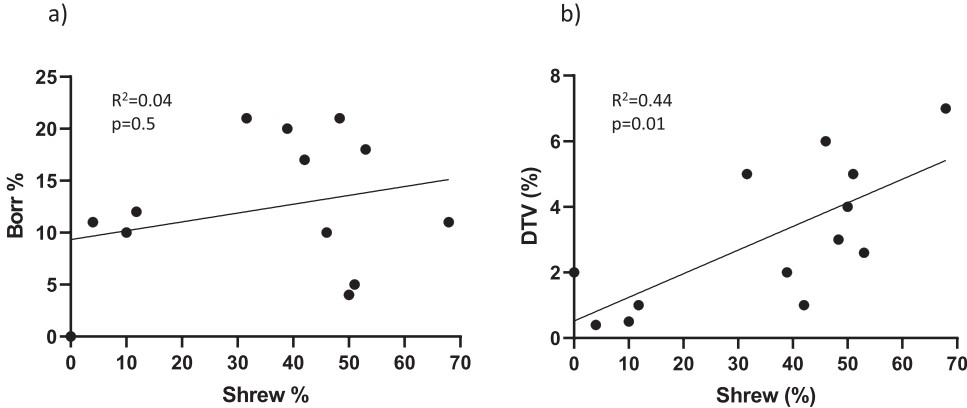

a)

b)

**Fig. 3 Correlation analysis of the percentage of ticks that fed on shrews compared to the percentage of infected ticks at our field sites.** The
*B. burgdorferi* (Borr) data are shown in panel **a** and deer tick virus (DTV) data are shown in panel **b**. The percentage of DTV ($n = 20$, $p = 0.01$), but not
*B. burgdorferi* ($n = 128$, $p = 0.5$), in ticks at a site is associated with the percentage of ticks that fed on shrews.

**Table 3 Bloodmeal host identified from deer tick virus-infected ticks from each study site.**

| Site | Year | Tick ID | Bloodmeal host |
|---|---|---|---|
| Naushon | 2018 | Nau18-N23 | Shrew |
| | 2020 | Nau20-N48 | Shrew |
| | | Nau20-N50 | Bird |
| Martha's Vineyard | | | |
| VH | 2019 | MV19-N137 | Shrew |
| | | MV19-N170 | Shrew |
| | | MV19-N157 | Unknown |
| Chil-1 | 2019 | MV19-N1129 | Cat |
| Chil-2 | 2020 | MV20-N249 | Shrew |
| Chil-3 | 2020 | MV20-N1173 | Unknown |
| Chap-1 | 2019 | MV19-N852 | Unknown |
| Chap-2 | 2020 | MV20-N1102 | Shrew |
| | | MV20-N1107 | Shrew |
| | | MV20-N1122 | Shrew |
| | | MV20-N1127 | Shrew |
| WT | 2019 | MV19-N1738 | *Sciuridae* |
| | 2020 | MV20-N247 | Shrew |
| AQ | 2020 | MV20-N850 | Shrew |
| Nantucket | 2020 | NN20-376 | Shrew |
| Rhode Island | | | |
| MB | 2020 | RI20-N302 | Shrew/deer |
| Trust | 2020 | RI20-N637 | Unknown |

or be more likely to simultaneously serve as host to nymphs and larvae (co-feeding), than the other mammals present in our study sites. Virus has been detected from xenodiagnostic ticks removed from skunks, raccoons and opossum in New York[18]. As with other tick-transmitted infections, contributions to the DTV enzootic cycle are likely to be dependent on local conditions and other hosts than shrews may contribute to maintenance. However, the association of shrews with DTV-infected ticks across multiple transmission seasons and across diverse sites, suggests that additional studies of shrews would be useful. Further investigations, including laboratory transmission studies are necessary to quantify the reservoir capacity of these hosts.

Shrews have not previously been suggested as reservoir hosts for DTV or POWV, but they appear to be competent reservoirs for the related TBE virus in Eurasia[19–21]. When DTV was identified, white-footed mice were considered the likely reservoir given that these rodents maintained the tick-transmitted agents of Lyme disease, babesiosis, and human granulocytic anaplasmosis[10,22]. Shrews were considered to be poorly infested by ticks and thus

were considered to have lower reservoir capacity for *B. burgdorferi* and *B. microti*[23]; this suggestion has been reconsidered[24,25]. Mammal surveys in DTV endemic sites have failed to detect virus or specific antibody in shrews[18]. Our use of host bloodmeal remnant analysis on infected ticks directly identified the source of the infecting animal reservoir without needing to extrapolate from indirect evidence such as comparative host density, tick infestation indices, and prevalence of pathogen exposure, and could be used to better understand the mode of perpetuation of other high consequence tick-borne pathogens such as the rickettsial agent of Rocky Mountain spotted fever, or those causing American tick-borne hemorrhagic fevers (Bourbon or Heartland virus).

## Materials and methods

**Tick samples**. Host-seeking nymphal black-legged ticks were collected by drag sampling during June and July of 2018–2020 as part of our on-going surveillance for tick-borne pathogens in northeastern United States. Study sites included four islands off the coast of Massachusetts (Naushon, Nantucket, and multiple sites on Martha's Vineyard) as well as two mainland sites in Washington County, Rhode Island, "Trust" and "MB" (Fig. 1). The study sites are comprised of mixed hardwood successional forests, with poison ivy, greenbriar, and bittersweet understory, except for Nantucket which has mixed shrubs (bayberry, highbush blueberry) and grasses, with poison ivy and greenbriar understory. Only the collections that yielded DTV-infected ticks are included in the analyses. Ticks were frozen immediately after collection. Ticks from sites with high rates of mouse-fed ticks from a previous study were included as comparison sites; Nantucket 2018, Robin's Island (off the coast of Long Island, New York) 2018 and 2019[11].

**Tick screening**. Ticks were homogenized individually in PBS, and then a portion of the homogenate was pooled in groups of six. In 2018 and 2019, homogenate pools were extracted using spin columns (Qiagen Inc., Hilden, Germany) following the DNA extraction protocol as suggested by the manufacturer with the exception that the RNase step is excluded. With the scarcity of spin column reagents during the COVID pandemic, RNA extractions in 2020 were conducted using 50 ul of QuickExtract (Lucigen Corp. Middleton, WI), as per manufacturer's instructions. Pools were screened for viral RNA with conventional RT-PCR using POW1 (TGGATGACAACAGAAGACATGC) and POW2 (GCTCTCTAGCTT-GAGCTCCCA) primers in 2018[10], and by real time PCR using POW9466-F1,F2 (ACCATAACAAACATGAAAGTCCAACT, CCATCACAAA-CATGAAAGTCCAACT) and POW9537-R1,R2 (TGAGTCTGCTGGTCCGAT-GAC, CTGTGAGTCAGCTGGTCCTATGAC) with FAM-labeled probe POW9453-MGB (CCTTCCATCATGCGGAT)[2] in 2019 and 2020. Positive pools were identified and the ticks from those pools were extracted and tested individually to identify the infected tick. The remaining ticks from negative pools were extracted individually using HOTSHOT[26]. All ticks were also screened individually for *B. burgdorferi* using a previously published real time PCR assay[27].

**Identification of DTV**. Because our screening assay detects both Powassan virus lineages, we had a portion of the *NS5* (POW1/2 primers) and *envelope* (POWDTVf ATGGTGTGCAAGAGAGACCA and POWDTVr ACAGTYTGGGCGA-CATCAAT primers[4]) genes sequenced (Genewiz Inc., Cambridge, MA) to determine the identity of our viral RNA. The resulting sequences were concantenated,

**Table 4 The percentage of ticks at each site that tested positive for having fed on either a shrew (*Soricidae*), mouse (*Peromyscus*), squirrel (*Sciuridae*), or all other hosts tested (*Odocoileus, Aves, Felis, Arvicolinae*, or *Lagomorpha*).**

| Site | Year | No. tested | Shrew | Mouse | Squirrel[a] | All other hosts |
|---|---|---|---|---|---|---|
| | | | No. pos (%) | No. pos (%) | No. pos (%) | No. pos (%) |
| Naushon | 2018 | 28 | 14 (50) | 1 (4) | 0 | 3 (11) |
| | 2020 | 28 | 19 (68) | 3 (11) | 0 | 5 (18) |
| Martha's Vineyard | | | | | | |
| VH | 2019 | 57 | 29 (51) | 2 (4) | 0 | 17 (30) |
| Chil-1 | 2019 | 47 | 0 | 6 (13) | 0 | 14 (30) |
| Chil-2 | 2020 | 38 | 20(53) | 6 (16) | 1 (3) | 12 (32) |
| Chil-3 | 2020 | 29 | 14 (48) | 1 (3) | 0 | 4(14) |
| Chap-1 | 2019 | nd[b] | — | — | — | — |
| Chap-2 | 2020 | 50 | 23 (46) | 6 (12) | 1 (2) | 6 (12) |
| WT | 2019 | 54 | 21 (39) | 7 (13) | 14 (26) | 15 (28) |
| | 2020 | 50 | 21 (42) | 2 (4) | 5 (10) | 5 (10) |
| AQ | 2020 | 19 | 6 (32) | 1 (5) | 0 | 9 (47) |
| Nantucket | 2020 | 51 | 6 (12) | 5 (10) | 2 (4) | 27 (53) |
| Rhode Island | | | | | | |
| MB | 2020 | 50 | 5 (10) | 1 (2) | 21 (42) | 7 (14) |
| Trust | 2020 | 50 | 2 (4) | 10 (20) | 2 (4) | 18 (36) |

[a]Primers amplify other *Sciuridae*.
[b]nd not done.

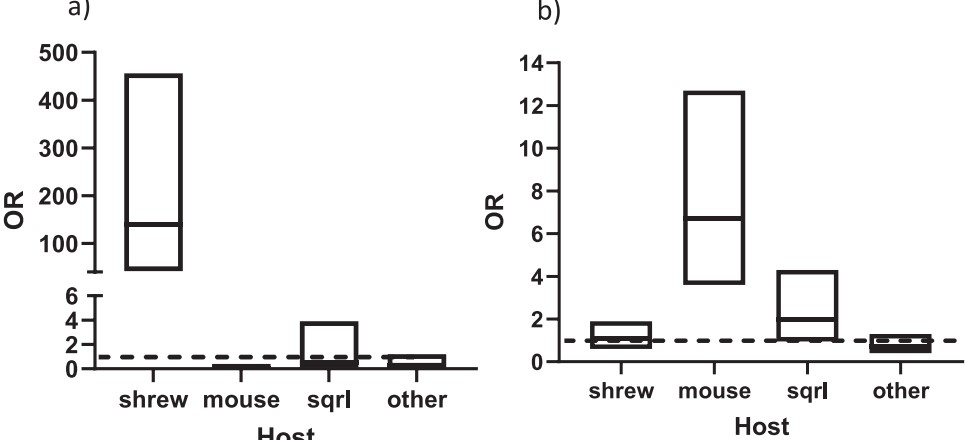

**Fig. 4 The likelihood that an infected tick had fed on either a shrew, mouse, squirrel (or other *Sciuridae*), or other host.** The data for deer tick virus-infected ticks are shown in panel **a**, and the data for *B. burgdorferi*-infected ticks are shown in panel **b**. Data are represented by boxplots of odds ratios (OR) with 95% confidence intervals, and all field sites are combined (n = 20 deer tick virus-infected ticks, n = 128 *B. burgdorferi*-infected ticks). A line is drawn at OR = 1, and any confidence interval that crosses it is not statistically significant. Sqrl= squirrel (or other *Sciuridae*).

aligned with verified DTV and POW sequences downloaded from GenBank using Geneious Prime 2020.1.2 (www.geneious.com), and trees were made using MEGAX[28]. Sequences from this study have been deposited into GenBank (Accession # MZ148230-MZ148271).

**Bloodmeal identification.** Approximately 50 ticks from each site (or all the ticks from a site if fewer than 50 were collected) as well as the DTV-infected ticks were tested for bloodmeal host identification using our existing assays targeting retro-transposons of mice, *Peromyscus leucopus*, and deer, *Odocoileus virginianus*[11]. Briefly, samples were tested in duplicate using ssofast qPCR master mixes (Bio-Rad Laboratories, Hercules, CA) for 50 cycles at 58 °C annealing temperature. New primers targeting voles, shrews, rabbits, squirrels/chipmunks/groundhogs, birds, and cat were designed and tested for sensitivity and specificity as previously described (Tables 1, 5)[11]. To do this, known retrotransposon motifs that were closest to the target host were downloaded from SineBase (https://sines.eimb.ru/). These were then used to search GenBank for similar sequences from the target host and other closely related species. If the target species have no matching sequences on GenBank, such as *Sylvilagus floridanus*, sequences from a related species were used instead. Primers and probes were designed using Geneious and tested against a panel of positive control DNA that included other potential hosts common in our field sites (including *Odocoileus virginianus, Peromyscus leucopus, Microtus*

*pennslyvanicus, Sylvilagus floridanus, Rattus rattus, Mephitis mephitis, Procyon lotor, Tamias striatus* etc (see ref. [11] for full panel) as well as unfed larval ticks to test for cross-reactions against tick DNA. Primers were redesigned multiple times to ensure adequate sensitivity and specificity. Specificity did not reach the species level for most primer sets; most were specific only to the family level (see Tables 1, 5). Previous work determined that a set of primers needed to detect positive control DNA diluted to a concentration of 10−7 ng/μl for optimal sensitivity in the assay. Primers were also tested with ticks that had been removed from known animals, including shrew, vole, chipmunk, rabbit and various bird species (Table 5). Knowing that these PCR assays are highly sensitive, similar to modern forensic methods, we followed strict pre- and post-PCR contamination prevention measures. These included: dedicated clean forceps for tick manipulations, physical separation of pre-PCR work areas from post-PCR work areas and the PCR machines, use of a PCR clean hood with UV decontamination between runs, and the inclusion of negative controls at each step of the process.

**Shrew samples.** Shrews were snap-trapped on Naushon and the WT site on Martha's Vineyard in September 2020. Three shrews were collected in 120 trap-nights. A sample of spleen and brain tissue was removed and RNA was extracted using Qiagen spin columns as suggested by the manufacturer. They were then tested for DTV as described above. The positive sample was verified by amplicon

**Table 5 Sensitivity of the bird primers.**

| Common name | Scientific name | MCZ accession number | Sensitivity (ng/µl) |
|---|---|---|---|
| Gray Catbird | *Dumetella carolinensis* | MCZ FN 09-116 | $10^{-7}$ |
| Eastern Towhee | *Pipilo erythrophthalmus* | MCZ FN 08-093 | $10^{-7}$ |
| Common Yellowthroat | *Geothlypis trichas* | MCZ FN 10-031 | $10^{-6}$ |
| Ovenbird | *Seiuru saurocapilla* | MCZ FN 12-100 | $10^{-6}$ |
| Carolina Wren | *Thryothorus ludovicianus* | MCZ FN 08-029 | $10^{-7}$ |
| Wood Thrush | *Hylocichla mustelina* | MCZ FN 12-042 | $10^{-7}$ |
| Swainson's Thrush | *Catharus ustulatus* | MCZ FN 07-044 | $10^{-7}$ |
| Hermit Thrush | *Catharus guttatus* | MCZ FN 17-1175 | $10^{-7}$ |
| American Robin | *Turdus migratorius* | MCZ FN 09-091 | $10^{-7}$ |
| Veery | *Catharus fuscescens* | MCZ FN 02-724 | $10^{-7}$ |
| House Wren | *Troglodytes aedon* | MCZ FN 09-101 | $10^{-6}$ |
| American Redstart | *Setophaga ruticilla* | MCZ FN 10-018 | $10^{-6}$ |

The top 12 bird species most commonly parasitized by *Ixodes* ticks as described by Halsey et al. metaanalysis[29] were tested with our bird primers using serial dilutions of positive control DNA. Specimens were obtained from the Harvard Museum of Comparative Zoology (MCZ). Sensitivity is determined by the last dilution (ng/µl) that tested positive.

sequencing as described above. Brain from this shrew was homogenized in buffered salt solution and clarified by low speed centrifugation. Isolation of live virus was attempted via suckling mouse inoculation. The supernatant was sterilized by filtration (0.22 micron) and 30 microliters intracerebrally inoculated into 5-day-old CD-1 mice in duplicate. The mice were held for 14 days and checked daily for evidence of neurologic disease. Our use of animals is covered under existing Tufts University IACUC approvals.

**Statistics and reproducibility**. Graphs were created using GraphPad Prism version 9 (www.graphpad.com) and correlation analysis was conducted using the analysis within that program. Odds Ratios and 95% confidence intervals were calculated using the online StatPages (https://statpages.info/ctab2x2.html). Blood-meal analysis PCRs were run in duplicate. If only one sample tested positive, the sample was rerun. Only samples that were reproducible were considered positive. Sample sizes for each field site can be found on Table 2.

**Reporting summary**. Further information on research design is available in the Nature Research Reporting Summary linked to this article.

## Data availability
Sequences from this study have been deposited into GenBank (Accession # MZ148230-MZ148271). All other data has been deposited in OSF and can be accessed at https://osf.io/5dzyp/?view_only=7b9dfd42baf2472aa802cb88efbc94d1. The map was adapted from an open source website (http://www.amaps.com/mapstoprint/LIST%20OF%20states.htm).

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

## Acknowledgements

Nantucket Conservation Foundation and the Naushon Trust granted research access to the Nantucket Field Station and Naushon properties, respectively. The Harvard Museum of Comparative Zoology loaned bird tissues used as positive control material for developing our bird assay. We thank these individuals and agencies. This is a contribution of the University of Massachusetts Nantucket Field Station and the Tufts Lyme Disease Initiative. We are supported by the National Institutes of Health grants R01 AI 130105 and AI 13742, the Rainwater Foundation, the Cedar Tree Foundation, Martha's Vineyard Vision Fellowship, and the Martha's Vineyard Tick-borne Illness Reduction Initiative.

## Author contributions

Conceptualization: H.K.G., S.R.T., Methodology: H.K.G., Investigation: H.K.G., S.R.T., T.N.M. and R.W.J., Funding acquisition: S.R.T. and R.W.J., Writing—original draft: H.K.G., Writing—review and editing: H.K.G. and S.R.T.

## Competing interests

The authors declare no competing interests.
