## [Transparent Peer Review File · Communications Biology]

Reviewers' comments:

Reviewer #1 (Remarks to the Author):

For tick-borne pathogens, many of which are transmitted by the nymphal tick stage, identifying the host on which the larval stage feeds provides important information on the natural history of the pathogen. The use of a retrotransposon targeted PCR assay to identify the host species source of the larval bloodmeal, as described in the manuscript, is therefore of considerable interest, especially when the host(s) involved in maintaining the tick-borne pathogen transmission cycle is unknown, as in the case of deer tick virus (DTV). Several factors need to be considered, which are elaborated below.

1) In the context of tick-borne pathogens, the reservoir host is usually described as the host that maintains the pathogen when conditions preclude active pathogen transmission. For example, a pathogen might persist in eg. a squirrel over winter and then become a source of infection for ticks in spring. Often ticks act as reservoir hosts for tick-borne pathogens because they frequently support infections for long periods and during adverse conditions. The only evidence that shrews are reservoir hosts was the detection of DTV in the brain of a shrew, suggesting a prolonged infection. However, only one shrew was virus positive and infectious virus was not isolated so the evidence is not strong. Furthermore, infection of the brain per se is a dead-end as far as a tick-borne pathogen is concerned.

2) Trans-stadial transmission is the persistence of a tick-borne infection from one tick stage to the next eg. infection of a larva which then persists through moulting to the nymph. In this context transmission is a controversial though commonly used term as the infection is not passing from one individual to another new individual and more accurately should be described as trans-stadial persistence. The authors claim their analysis "incriminates transstadial transmission between shrews and larval deer ticks" (Line 114). This does not make sense as "transstadial" means from one stadium (eg. larva) to the next (nymph). Horizontal transmission is a more appropriate term, meaning from one host (shrew) to another host (larva).

3) Although the authors reject the hypothesis DTV was inherited by infected nymphs, they have not excluded the possibility of transovarial transmission to larvae followed by amplification of the infection through co-feeding larvae. This hypothesis has been postulated for TBEV. Studies with TBEV have also shown that co-feeding (non-systemic/non-viremic) transmission only occurs on certain susceptible hosts (infection of the host occurs even though the host is seemingly non-viremic). Thus the authors have not excluded the possibility shrews are more effective than mice in supporting co-feeding transmission.

Specific comments:

Line 62: define "convenience samples"

Line 148-149: It may have helped to include data from collections that did not yield DTV-infected ticks. For example, if these collections showed little evidence larvae were feeding on shrews, the results could perhaps support the implication of shrews in the maintenance of DTV.

Line 193: arbovirus isolation by intracerebral inoculation of newborn mice was once the method of choice. Mice 5 days old tend to have lost their exquisite susceptibility to arboviruses.

Reviewer #2 (Remarks to the Author):

See attached review

Reviewer #3 (Remarks to the Author):

The authors describe their attempt to determine the wildlife species able to transmit Powassan Virus Lineage II (aka DTV) to blacklegged ticks, *Ixodes scapularis*. By my count, they tested 20 nymph-stage ticks that were positive for DTV, using a retrotransposon PCR assay that detects remnants of host nucleic acid from prior blood meals. Almost all of the prior blood meal is digested and

metabolized or converted into tick tissue, but the minute remnants are apparently sufficient to be identified by the assay apparently developed by these authors (ref 11). Of the 20 DTV-positive ticks, the putative blood meal host was a "bird" (N=1), "cat" (N=1), shrew/deer (N=1), squirrel (N=1), unknown (N=4), or a "shrew" (N=12). Shrews comprised roughly half of all hosts (using the blood meal assay), irrespective of infection with DTV. The percentage of ticks identified as having fed previously on shrews correlated positively with the percentage of ticks positive for DTV. These observations lead the authors to conclude that shrews are *the* reservoir host for DTV in blacklegged ticks.

Identifying the species (plural) of host that contribute to zoonotic transmission is an important undertaking. Although it remains quite rare, Powassan viral meningoencephalitis is of public health concern, given the possible increasing trends in prevalence and adverse health outcomes. Basic information on the transmission cycle between wildlife hosts and blacklegged tick vectors will contribute to both basic and applied science of this disease. Of course, the more specific the information (e.g., the species, rather than the families or orders of hosts), the more useful it will be.

I have three major concerns with this paper and several more minor concerns. All but the first one should be straightforward to correct, and the first concern is addressable with modest additional laboratory effort. The major concerns are as follows:

1. The authors rely on a prior publication (ref 11) for describing the blood meal assay upon which this paper fundamentally depends. Ref 11 is well-detailed and seems reliable. However, that paper describes the fundamentally important ground-truthing of the retrotransposon assay by testing the molecular method against ticks known to have fed from a particular species of host (in other words, ticks that were directly collected from that species of host). This was conducted in ref 11 for *Peromyscus leucopus*. In the present paper, no such ground-truthing is described, and it seems it was not performed. Therefore, the molecular assay for other hosts, including all those in this paper, does not appear to have been confirmed. I think it is necessary to collect ticks known to have fed from shrews, and hopefully other hosts, and determine the accuracy of the molecular method against this gold standard.
2. The authors unnecessarily and inaccurately fall prey to the common thinking that there is a single reservoir host ("the" reservoir host) for any given zoonotic pathogen. Of course, this is rarely the case, and particularly not an accurate depiction of the known blacklegged tick-borne pathogens. Typically, several reservoir hosts are capable of transmitting any given tick-borne pathogen (and sometimes there are many), although they differ quantitatively in what's called reservoir competence. Indeed, the author's own data reveal at least three other reservoir taxa (discounting the "shrew/deer"). And prior studies, some of which are cited, provide direct evidence of other reservoir hosts for DTV. Hence is it simply inaccurate to refer to "the" reservoir, and this should be corrected throughout the ms to refer to "an important" reservoir, or some other similar wording that makes the lack of singularity clear. Such a change in tone is all the more important given that the assay is not species-specific, nor even genus-specific. As the authors indicate, the species identity or identities of the shrews involved is unknown, as is the case with the birds and squirrels (although the "cat" is likely a house cat).
3. The authors use a discredited nomenclature for the tick species they are studying. The Entomological Society of America is the accepted authority on both Latin binomials and common ("trivial") names for arthropods. This source lists only the binomial *Ixodes scapularis* and the common name blacklegged tick for this species. The reasons for having only a single accepted scientific name and common name are clear and have been established since Linnaeus – it removes a key source of ambiguity in identifying species. "Deer tick" and "Ixodes dammini" are technically illegitimate names, not accepted as valid by taxonomic nomenclature. This needs to be changed throughout.

Minor points:

1. Lines 36-37: When stating, "All tick-borne encephalitis cases in North America have historically been attributed to POWV...", what do the authors mean by "historically"? Clearly, many cases of DTV causing encephalitis have been documented. Please be more precise.

2. Line 39: DTV is not a zoonosis. It is the agent causing the zoonosis, which itself is the meningoencephalitis.
3. Sentence starting line 54: This passage makes no sense. Why would one assume that a highly catholic (generalized) tick vector would acquire all zoonotic pathogens from a single host species?
4. Sentence starting line 74: This statement about a lack of association between *B. burgdorferi* infection and DTV infection requires statistical support.
5. Line 79-80: It is somewhat concerning that one of the ticks was identified as having fed on multiple hosts (shrew and deer), given that the larval stage feeds only once. What was the likely cause of this designation?
6. Section starting line 100: The observation that most of the infected ticks could be assigned to a host taxon in no way allows the "rejection" of the hypothesis that DTV is transovarially transmitted, so this statement must be changed. The small sample of ticks, combined with the roughly 20% "unknown" sources of infection, means that some DTV positives certainly could have arisen from vertical transmission, for which there is evidence from closely related TBEV.
7. Figure 2 and associated statistics: These associations would appear to be better represented by correlation rather than regression analyses. The basic results will not change, but the approach will be more appropriate, given the lack of a clear independent variable.

Reviewers' comments:

Reviewer #1 (Remarks to the Author):

For tick-borne pathogens, many of which are transmitted by the nymphal tick stage, identifying the host on which the larval stage feeds provides important information on the natural history of the pathogen. The use of a retrotransposon targeted PCR assay to identify the host species source of the larval bloodmeal, as described in the manuscript, is therefore of considerable interest, especially when the host(s) involved in maintaining the tick-borne pathogen transmission cycle is unknown, as in the case of deer tick virus (DTV). Several factors need to be considered, which are elaborated below.

1) In the context of tick-borne pathogens, the reservoir host is usually described as the host that maintains the pathogen when conditions preclude active pathogen transmission. For example, a pathogen might persist in eg. a squirrel over winter and then become a source of infection for ticks in spring. Often ticks act as reservoir hosts for tick-borne pathogens because they frequently support infections for long periods and during adverse conditions. The only evidence that shrews are reservoir hosts was the detection of DTV in the brain of a shrew, suggesting a prolonged infection. However, only one shrew was virus positive and infectious virus was not isolated so the evidence is not strong. Furthermore, infection of the brain per se is a dead-end as far as a tick-borne pathogen is concerned.

Modes of perpetuation of the TBE complex of viruses have a long conceptual history. The original investigations of RSSE by Silber and Soloviev (1946) used the term "reservoir" to denote the host in which virus persisted for long durations (either tick or persistently infected vertebrate), as does Reviewer #1. Silber and Soloviev, and Pavlovsky as well, thought the tick to be the main reservoir. Smith et al. 1964 used the term amplifying host for louping ill virus maintenance. McLean et al. 1962 adopted Hess and Holden 1958's definition of reservoir ("a host that serves as a source of infection for other hosts, including vectors") in his studies of Powassan virus ecology. Nosek and Grulich 1967 used the term in what is thought of (at least by Americans) as the classical sense, with vertebrate hosts serving as a source of virus to ticks that passed it to other hosts, but distinguished long term reservoirs (persistent infection) such as hedgehogs from those that are transiently infectious such as rodents. Pavlovsky referred to donors and recipients, a similar concept. Hartemink et al 2008 eschewed the term reservoir, referring only to competent hosts for systemic (classical horizontal transmission from viremic animal to tick) and nonsystemic transmission (cofeeding); their model concluded that cofeeding was the most influential to perpetuation, but that systemic transmission contributed depending on local circumstances

and hosts; transovarial transmission was thought to play a minor role. Michelitsch et al. 2019 provides a nice review of the three modes of perpetuation for TBEV, and explicitly suggests the critical need for a reservoir that serves as amplifying host, such as the bank vole. An amplifying host could allow either systemic or nonsystemic transmission. Because the term "reservoir" can mean different things to different researchers, we now define the term at the very beginning of our paper (lines 53-54), clarifying that we view the term reservoir to mean "amplifying host". In our original manuscript, we were careful to include the possibility of shrews contributing to either systemic or nonsystemic transmission (lines 133-135); both modes would amplify prevalence of virus in host seeking ticks.

If ticks identified in this study had acquired infection transovarially and therefore had emerged from the egg already infected, the infection status of the tick would have no association with any mammalian host. It simply would not matter which host the tick had fed on, and the hosts identified in infected ticks would be representative of the hosts present at any particular site. We would have expected to detect more mice and deer as bloodmeal hosts, because these animals are more likely to serve as hosts for immature deer ticks at many sites (see our previous bloodmeal analysis paper). Furthermore, there should be no association between infection prevalence and the dominant bloodmeal host at our sites; that is, we should be just as likely to have detected infected ticks in sites dominated by mice, deer, shrew or any other mammalian host. But this was not the case. Therefore, we do not believe that the majority of the ticks in our study were infected through transovarial transmission. This argument is presented in lines 108-110.

Our data shows that shrews were, by far, the most likely host identified in DTV-infected ticks. Furthermore, field sites with high numbers of ticks feeding on shrews were associated with higher rates of infection in ticks. This indicates that shrews play a role in DTV transmission at a site, likely as a source of virus for larval ticks feeding on them.

We agree that virus in the brain of a shrew is not likely to be infectious by tick bite, but it does show that the shrew is capable of being infected and the viremia was systemic at some point before infecting the brain tissue. Demonstrating the virus or its cognate sequence is better evidence for the possibility of host infectivity than the presence of antibody.

2) Trans-stadial transmission is the persistence of a tick-borne infection from one tick stage to the next eg. infection of a larva which then persists through moulting to the nymph. In this context transmission is a controversial though commonly used term as the infection is not passing from one individual to another new individual and more accurately should be

described as trans-stadial persistence. The authors claim their analysis “incriminates transstadial transmission between shrews and larval deer ticks” (Line 114). This does not make sense as “transstadial” means from one stadium (eg. larva) to the next (nymph). Horizontal transmission is a more appropriate term, meaning from one host (shrew) to another host (larva).

We agree that the term transstadial transmission is used too loosely by diverse researchers, including us, but it is a term commonly used in work on the ecology of tick borne infections for when a tick acquires the agent from a mammalian host as a larvae and maintains the infection through the molt to the nymphal stage. The ticks in our study were most likely infected during the larval bloodmeal, and we tested them after they molted to nymphs; so they have indeed undergone transstadial transmission or “persistence” through the molt. In the spirit of trying to change the imprecision in the literature, we have changed to horizontal transmission throughout the manuscript as suggested by the reviewer.

3) Although the authors reject the hypothesis DTV was inherited by infected nymphs, they have not excluded the possibility of transovarial transmission to larvae followed by amplification of the infection through co-feeding larvae. This hypothesis has been postulated for TBEV. Studies with TBEV have also shown that co-feeding (non-systemic/non-viremic) transmission only occurs on certain susceptible hosts (infection of the host occurs even though the host is seemingly non-viremic). Thus the authors have not excluded the possibility shrews are more effective than mice in supporting co-feeding transmission.

We agree. We state this starting on line 124

“Shrews are thus the main candidate for the vertebrate DTV reservoir but we cannot now rank the contribution of transstadial transmission relative to other modes of perpetuation. Shrews may be more likely to sustain an infectious viremia, or be more likely to simultaneously serve as host to nymphs and larvae (cofeeding), than the other mammals present in our study sites.”

Regardless of mode of transmission, we find it notable that infected nymphs had more frequently fed on shrews as larvae. This is a strong hint that shrews contribute to the perpetuation of DTV in nature, at least in our sites, and further investigations should be conducted on them.

Specific comments:

Line 62: define "convenience samples"

The term convenience sample in epidemiology denotes a non-probabilistic sampling method, i.e., non random. We used "convenience samples" because they come from any site where we had sufficient ticks and the sites were known to be enzootic for the other deer tick-transmitted infections. There was no a priori sampling design and the sampling efforts were not strictly controlled to be similar among the sites.

Line 148-149: It may have helped to include data from collections that did not yield DTV-infected ticks. For example, if these collections showed little evidence larvae were feeding on shrews, the results could perhaps support the implication of shrews in the maintenance of DTV.

We plan to present more complete data on bloodmeal identification from various sites in a separate publication and did not want to present that data here. However, we have already published mouse and deer bloodmeal data from Nantucket, Rhode Island, and Robin's Island from 2018 and 2019 (see Goethert 2021). We have tested the ticks from sites with high rates of mouse-fed ticks for DTV and failed to identify any positives. We have added this to the manuscript (Lines 87-90, and 161-162).

Line 193: arbovirus isolation by intracerebral inoculation of newborn mice was once the method of choice. Mice 5 days old tend to have lost their exquisite susceptibility to arboviruses.

We recognize the susceptibility issue but sadly this age was all that was available at the time of analyzing the shrew brains (material was held refrigerated and inoculated as soon as possible (within 48 hours) after being identified as positive by RT-PCR). Freezing brain tissue does lead to some loss of infectivity, and we chose to attempt isolation with nonfrozen materials. It should be noted that we routinely isolate DTV from ticks using 5 day old mice, although we have not directly determined whether we would be more successful using younger mice.

Reviewer #2 (Remarks to the Author):

Goethert et al. Shrew reservoir host for Powassan virus

In this field and laboratory study, the likely reservoir host for Powassan virus is explored with some novel bloodfeeding analyses and assessments. Overall, the study is novel and very interesting; a great addition to the literature. There are some significant changes, any editorial, that need to be completed before it is ready for publication.

Suggest changing the title to “Shrew likely reservoir host for Powassan virus” or similar. The low sample size was concerning to make a definitive statement and this was not validated with challenge studies.

done

L. 16 Remove extra space before “we” done

L. 17 Change to host-specific and white-footed mouse. Change throughout. done

L. 18. Change to “mice, considered the primary eastern...” done

L.19. change to “host-seeking” change throughout the text. See above

L. 19. Change “deer ticks” to “blacklegged ticks” throughout. Blacklegged tick is the correct common name and eliminates confusion. See next comment

L. 44. The use of *Ixodes dammini* here is not correct. According to Sanders *I. dammini* was synonymized with *I. scapularis* in 1993 by Oliver et al. and redescribed in 1996 to reduce confusion regarding identification. The use of *I. dammini* is a junior synonym and creates confusion for researchers, students, and public health officials.

We continue to use the junior subjective synonym as allowed by the International Code of Zoological Nomenclature, Article 23.3.6. “The Principle of Priority continues to apply to an available name when treated as a junior synonym; it may be used as the valid name of a taxon by an author who considers the synonymy to be erroneous, or if the senior synonym is found to be unavailable or invalid”. We consider the synonymy to be erroneous, and indeed the evidence continues to accumulate that what is called “*I. scapularis*” is likely a species complex. Although much is made about the hybridization experiments in Oliver et al. 1993, recent studies suggest that the formation of fertile F1 hybrids is not a good criterion for distinguishing between *Ixodes* species (Kovalev et al 2015, demonstrating natural fertile hybrids of *I. persulcatus* and *I. pavlovskyi*). It is very likely that the synonymy will be reversed in the near future. In today’s world of easy access to information, it is unlikely that the use of *I. dammini* or deer tick confuses many with at least a working knowledge of Lyme disease epidemiology. (In reality, deer tick is more commonly used than blacklegged tick by people in the northeastern U.S.) We accordingly thank the reviewer for his or her opinion but continue to use *I. dammini* and the colloquial deer tick; we clearly mention in our paper (line 45) that *I. dammini* is a junior subjective synonym. We fail to see how the nomenclature influences the scientific merit of our findings. We believe we have provided a finding that may lead to better understanding the natural history of a relatively poorly studied but emerging high consequence tick borne infection. The nomenclature should play no role in evaluating the merit of the work.

L. 57 More than *Peromyscus* is a reservoir (e.g., *Peromyscus maniculatus* and *Tamias striatus*, etc.) We agree that there are more reservoir hosts, but *P. leucopus* is considered the most important. We have added the word “primarily” to make this point more clear.

L. 58. Add “wild” before “mice and delete “from the field” done

L. 73 & L.74. Check formatting guidelines, but it is generally not convention to start a sentence with an abbreviated word (e.g., *B. burgdorferi*) fixed

L.85. Space needed between median 10.5 fixed

L86 Check formatting, but convention is to spell out all not measurement numbers under 10. Change throughout. fixed

L. 116. How did you differentiate between species? We didn't. We state that our bloodmeal analysis does not differentiate between shrew species (see table 1). We added more details in the methods (Line 203)

Overall – Mice and shrew generalized isn't informative. Try to add species names where you can. We added on line 58 that when we refer to mouse, we are referring to *P. leucopus*. We added the species names to our methods section (line 176 to clarify). However, our assays are not species specific, and we believe that using the common names makes the manuscript more readable; higher order scientific names are usually less familiar to the general reader.

L. 85. What squirrel species? As our assay does not differentiate between squirrels, chipmunks and other *Sciuridae*, therefore we do not specify a species here. There are multiple species of squirrels native to the area, and we cannot differentiate between them.

Why were chipmunks not investigated? Make sure to include why it wasn't included.

Chipmunks were included within our squirrel primers. As we have said above, the primers pick-up chipmunks along with squirrels. Although when we originally designed the primers, we intended them to be specific for squirrels, but as we say on line 181 in the methods, they do not differentiate between squirrels, chipmunks and groundhogs. We have removed “squirrel” from our bloodmeal identification in Table 3 and left just *Sciuridae* to emphasize that we cannot know whether it is a squirrel, a chipmunk, a groundhog etc..

L. 142 At the end of the section, add that laboratory validation studies are needed, and make sure to list study limitations including the potential or co-feeding contamination and this being correlation, not causation. Also the limited shrew samples.

The last paragraph ending with line 142 emphasizes the more general use of our analysis.

In the previous paragraph (lines 119-133), we summarize our supporting data and state our conclusions, and I believe that we touched on most of the points that the reviewer was looking for. We state on line 124 “the positive association between the proportion of shrew-fed ticks and the prevalence of DTV infection”. We take care to use the word “association” to denote that we do not have definitive evidence. Further in that paragraph line 126, we state, “Shrews are thus the main candidate for the vertebrate DTV reservoir”. Once again emphasizing that our data is not conclusive, but points to shrews as a candidate for further investigation. Finally, we state on line 128, “Shrews may be more likely to sustain an infectious viremia, or be more likely

to simultaneously serve as host to nymphs and larvae (cofeeding), than the other mammals present in our study sites.” thus reiterating that we cannot differentiate between co-feeding and transstadial transmission.

We have added reference to our small sample size on line 122. We note that the finding of virus-infected ticks having fed more frequently on shrews as larvae would suffice for this report; we were gratified that a quickly obtained limited small sample of shrews yielded evidence of active infection from a site where virus infected ticks were collected. In the near future, we will be trying to obtain additional shrews with targeted sampling methods (e.g., pitfall traps).

We have added a final sentence on line 131 to emphasize that further investigations are needed. *“ Further investigations, including laboratory transmission studies are necessary to quantify the reservoir capacity of these hosts.”*

There needs to be more detail in the methods
We expanded our methods section (Lines 194-208).

L. 148. The site names are not helpful. A map would be useful. **We have added a map (Fig. 1)**
Line 340.

Fig. 1. Map of the field sites included in this study.

How were ticks stored after collection? They were frozen immediately after arrival in the laboratory.

L. 161 what is the manufacturer of HOTSHOT? This is not a commercial kit. We have added the reference.

L. 162 Should *Borrelia burgdorferi* be abbreviated here? OK- changed to *B. burgdorferi*
What were the study sites like? Urban? Suburban? Forested? Give some description.

The study sites comprise those that are known to be highly enzootic for the deer tick-maintained guild of infections (Lyme and others). Naushon Island, Chilmark MA, and Washington County RI sites are mixed hardwood successional forests, with poison ivy, greenbriar and bittersweet understory. The Nantucket sampling site comprises mixed shrubs (bayberry, highbush blueberry) and grasses, with poison ivy and greenbriar understory. For all sites, ticks were sampled by randomly dragging or sweeping the understory, or along edges of paths (including deer trails). We have added this to the methods (Line 152).

L. 166 remove extra parentheses One is for the reference and one is for the statement

L 181 Add a hyphen after pre- and post- OK

L. 187 How were shrews trapped? Were they alive? What baits were used? What traps were used?

Shrews were trapped using museum special snap traps baited with peanut butter and set in lines, 10m apart. Traps were set late in the day and checked early in the morning. Captures (kills) were immediately placed on blue ice packs and held refrigerated until their necropsy, usually within 24 hours. Details were added to the methods (Line 209).

L. 193. Change to 5-day-old or 5-d-old depending on style guidelines. Why were mice so young? Suckling mouse inoculation is the standard practice for isolating encephalitic flaviviruses.

L194 to reduce confusion, suggest giving the species of mouse used. We specify CD-1 mice in the text (Line 216).

I couldn't find the relevance of the use of these mice. Where was this described in the text?

Needs to be made more clear why this was done and the results. (Line 207) Added a sentence that specifies that the mouse inoculation was done to attempt to isolate live virus.

We state the results on (Line 99), "Attempts to isolate virus by suckling mouse inoculation failed." Not sure what more we can say about this.

What is the IACUC protocol number? Tufts University, G2020-101, expiration 2/2023.

Figures and tables in general need to stand alone. There are several cases where the captions are not descriptive. The tick species and animal species should be given for clarity
Figure 2. DTV should be spelled out in the captions so they can stand alone. Scientific names of shrew(s) need to be added as well as the tick species.

DTV has been spelled out in all figure and table captions. Scientific names of shrews are not used because we cannot be certain which species it is.

Figure 1. Positive sample of what? Spell out DTV and POW Sample was changed to “tick” and DTV and POW were spelled out.

Figure 4. Spell out sqrl or define in legend. Use scientific name or add “generalized” to genus or similar.

Sqrl is defined in the figure legend and the scientific name is added.

Table 1. Add info on scientific name, or if you don’t have a scientific name of a specific species, use genus or higher taxonomic group. Table title needs more details to stand alone.

We have changed the table headings for added clarity. We kept the common names in the first column to correspond with the common names we use throughout the paper. The scientific names of the genus or family that each primer set picks up is listed in the last column. More details have been added to the footnotes.

Table 2. Table title needs to stand alone. Added more details.

Table 3. Table title needs to stand alone. Use scientific name or define species/genus in footnotes. As we have said above, because the genus and species of shrew or bird are unknown, the scientific names are not included in the table. We have changed squirrel to Sciuridae to emphasize that it could be a squirrel, chipmunk or groundhog. Table title was changed for clarity.

Table 5. Should be “gray” or “Grey” catbird (both are correct), and “Common” yellowthroat. Spacing of scientific names is wrong. Revise the table title so “using” is in the same sentence twice. Done

Table 4. The title should stand alone. “nd” needs to be explained in the footnotes. All other hosts need to be defined in the footnotes. Scientific name need to be used in column titles.

Added scientific names to the table title to clarify.

Reviewer #3 (Remarks to the Author):

The authors describe their attempt to determine the wildlife species able to transmit Powassan Virus Lineage II (aka DTV) to blacklegged ticks, Ixodes scapularis. By my count, they tested 20 nymph-stage ticks that were positive for DTV, using a retrotransposon PCR assay that detects remnants of host nucleic acid from prior blood meals. Almost all of the prior blood meal is digested and metabolized or converted into tick tissue, but the minute remnants are apparently sufficient to be identified by the assay apparently developed by

these authors (ref 11). Of the 20 DTV-positive ticks, the putative blood meal host was a "bird" (N=1), "cat" (N=1), shrew/deer (N=1), squirrel (N=1), unknown (N=4), or a "shrew" (N=12). Shrews comprised roughly half of all hosts (using the blood meal assay), irrespective of infection with DTV. The percentage of ticks identified as having fed previously on shrews correlated positively with the percentage of ticks positive for DTV. These observations lead the authors to conclude that shrews are *the* reservoir host for DTV in blacklegged ticks.

Identifying the species (plural) of host that contribute to zoonotic transmission is an important undertaking. Although it remains quite rare, Powassan viral meningoencephalitis is of public health concern, given the possible increasing trends in prevalence and adverse health outcomes. Basic information on the transmission cycle between wildlife hosts and blacklegged tick vectors will contribute to both basic and applied science of this disease. Of course, the more specific the information (e.g., the species, rather than the families or orders of hosts), the more useful it will be.

I have three major concerns with this paper and several more minor concerns. All but the first one should be straightforward to correct, and the first concern is addressable with modest additional laboratory effort. The major concerns are as follows:

1. The authors rely on a prior publication (ref 11) for describing the blood meal assay upon which this paper fundamentally depends. Ref 11 is well-detailed and seems reliable. However, that paper describes the fundamentally important ground-truthing of the retrotransposon assay by testing the molecular method against ticks known to have fed from a particular species of host (in other words, ticks that were directly collected from that species of host). This was conducted in ref 11 for *Peromyscus leucopus*. In the present paper, no such ground-truthing is described, and it seems it was not performed. Therefore, the molecular assay for other hosts, including all those in this paper, does not appear to have been confirmed. I think it is necessary to collect ticks known to have fed from shrews, and hopefully other hosts, and determine the accuracy of the molecular method against this gold standard.

To comply with the reviewer's recommendation, we have tested ticks directly removed from voles, chipmunks, shrews, rabbits, as well as various bird species and confirmed that our primers were able to successfully identify each. We have added this to the manuscript (Line 201).

2. The authors unnecessarily and inaccurately fall prey to the common thinking that there is a single reservoir host ("the" reservoir host) for any given zoonotic pathogen. Of course, this is rarely the case, and particularly not an accurate depiction of the known blacklegged tick-borne pathogens. Typically, several reservoir hosts are capable of transmitting any given tick-borne pathogen (and sometimes there are many), although they differ quantitatively in what's called reservoir competence. Indeed, the author's own data reveal at least three other reservoir taxa (discounting the "shrew/deer"). And prior studies, some of which are cited, provide direct evidence of other reservoir hosts for DTV. Hence is it simply inaccurate to refer to "the" reservoir, and this should be corrected throughout the ms to refer to "an important" reservoir, or some other similar wording that makes the lack of singularity clear. Such a change in tone is all the more important given that the assay is not species-specific, nor even genus-specific. As the authors indicate, the species identity or identities of the shrews involved is unknown, as is the case with the birds and squirrels (although the "cat" is likely a house cat).

We appreciate that there is almost never a sole reservoir host for most vector-borne infections, but we do not think that the evidence (some of which is our own work) incriminating *Peromyscus* as a reservoir host for DTV is convincing. Serological studies have suggested exposure of diverse mammals to Powassan virus, also in sites where the DTV lineage is known to be circulating. In the TBEV literature, seropositivity is simply interpreted as evidence for virus being actively transmitted in the site (e.g., Nosek and Grulich 1967) and not necessarily an indication of reservoir capacity. The classic studies of McLean and colleagues in the original Powassan investigations in the 1950s and 1960s which yielded viral isolates from mammals (mainly squirrels) were focused on what we now know is lineage I (prototype POW). We have long argued that the perpetuation of lineage I differs from lineage II (DTV). We regret if we gave the impression that we believe shrews to be the one and only reservoir host for DTV. There are likely to be others. We changed the wording to be more clear. (see title change as well as throughout manuscript referring to "a" reservoir host instead of "the" reservoir host.)

Also note that we do not believe that our study yields enough evidence to point to three other reservoir hosts. Since each of the other species (bird, cat and squirrel/chipmunk/groundhog) only had a single positive tick, we cannot make any conclusions regarding their role as reservoirs. It may be that these ticks were infected transovarially. Shrews were found to be associated with virus in multiple sites; although we

cannot exclude transovarial transmission as the source, it would be surprising if infected larvae were mainly feeding on shrews across our sites.

3. The authors use a discredited nomenclature for the tick species they are studying. The Entomological Society of America is the accepted authority on both Latin binomials and common (“trivial”) names for arthropods. This source lists only the binomial *Ixodes scapularis* and the common name blacklegged tick for this species. The reasons for having only a single accepted scientific name and common name are clear and have been established since Linnaeus – it removes a key source of ambiguity in identifying species. “Deer tick” and “*Ixodes dammini*” are technically illegitimate names, not accepted as valid by taxonomic nomenclature. This needs to be changed throughout.

We continue to use the junior subjective synonym as allowed by the International Code of Zoological Nomenclature (the accepted authority for animal nomenclature), Article 23.3.6. “The Principle of Priority continues to apply to an available name when treated as a junior synonym; it may be used as the valid name of a taxon by an author who considers the synonymy to be erroneous, or if the senior synonym is found to be unavailable or invalid”. We consider the synonymy to be erroneous, and indeed the evidence continues to accumulate that what is called “*I. scapularis*” is likely a species complex. Although much is made about the hybridization experiments in Oliver et al. 1993, recent studies suggest that the formation of fertile F1 hybrids is not a good criterion for distinguishing between *Ixodes* species (Kovalev et al 2015, demonstrating natural fertile hybrids of *I. persulcatus* and *I. pavlovskyi*). It is very likely that the synonymy will be reversed in the near future. In today’s world of easy access to information, it is unlikely that the use of *I. dammini* or deer tick confuses many with at least a working knowledge of Lyme disease epidemiology. Deer tick is more commonly used than blacklegged tick by the public as well as scientists in the northeastern U.S., ESA’s lists notwithstanding; a common name is simply one that is commonly used. (In some parts of the U.S., the deer tick is known as the bear tick.) We accordingly thank the reviewer for his or her comments but continue to use *I. dammini* and the colloquial deer tick; we clearly mention in our paper (line 44) that *I. dammini* is a junior subjective synonym. We fail to see how the nomenclature influences the scientific merit of our findings. We believe we have provided a finding that may lead to better understanding the natural history of a relatively poorly studied but emerging high consequence tick borne infection. The nomenclature should play no role in evaluating the merit of the work.

Minor points:

1. Lines 36-37: When stating, "All tick-borne encephalitis cases in North America have historically been attributed to POWV...", what do the authors mean by "historically"? Clearly, many cases of DTV causing encephalitis have been documented. Please be more precise.

We have changed this to be more specific . We (Line 37) meant that all cases before 2009 when El Khoury et al published the first definitive case of encephalitis caused by DTV.

2. Line 39: DTV is not a zoonosis. It is the agent causing the zoonosis, which itself is the meningoencephalitis.

Reworded this. "DTV was demonstrated as a human pathogen when its...."

3. Sentence starting line 54: This passage makes no sense. Why would one assume that a highly catholic (generalized) tick vector would acquire all zoonotic pathogens from a single host species?

Reworded this. "Because the deer tick is the vector for the agents of Lyme disease (*Borrelia burgdorferi*), human babesiosis (*Babesia microti*), and granulocytic anaplasmosis (*Anaplasma phagocytophilum*), and they all share the white-footed mouse (*Peromyscus leucopus*, hereafter referred to as mouse) as a primary reservoir host, the DTV reservoir has been assumed to be identical."

4. Sentence starting line 74: This statement about a lack of association between *B. burgdorferi* infection and DTV infection requires statistical support.

The p-value was added as well as reference to Figure 2 (Line 78).

5. Line 79-80: It is somewhat concerning that one of the ticks was identified as having fed on multiple hosts (shrew and deer), given that the larval stage feeds only once. What was the likely cause of this designation?

We do not believe that this is concerning at all. It is well known that ticks can be groomed off and are likely to reattach. (See Tahir et al 2020 for a recent review on this phenomenon [10.3390/microorganisms8060910](https://doi.org/10.3390/microorganisms8060910)). We regularly collect questing ticks that are visibly distended and partially fed. What is not known is how often this occurs in nature.

6. Section starting line 100: The observation that most of the infected ticks could be assigned to a host taxon in no way allows the “rejection” of the hypothesis that DTV is transovarially transmitted, so this statement must be changed. The small sample of ticks, combined with the roughly 20% “unknown” sources of infection, means that some DTV positives certainly could have arisen from vertical transmission, for which there is evidence from closely related TBEV.

We agree with the reviewer here and have reworded the sentence (Line 106).“ We found a clear association of DTV infection with a specific host from all study sites, indicating that these ticks were not likely to have inherited the infection. “

7. Figure 2 and associated statistics: These associations would appear to be better represented by correlation rather than regression analyses. The basic results will not change, but the approach will be more appropriate, given the lack of a clear independent variable.

We had done both with our original analysis but the regression analysis gave a better visual (the linear regression line) so that was what ended up in the paper. We have changed the figure; we left regression line and replaced the R^2 and p -values with those from the correlation analysis.

(A.)

(B.)

Fig. 3. Correlation analysis of the percentage of ticks that fed on shrews compared to the percentage of *Ixodes* ticks infected with *B. burgdorferi* (Borr) (A) or deer tick virus (DTV) (B). The percentage of DTV ($p=0.01$), but not *B. burgdorferi* ($p=0.5$), in ticks at a site is associated with the percentage of ticks that fed on shrews.

Reviewers' comments:

Reviewer #1 (Remarks to the Author):

Thank you for your constructive responses to the comments. I just have a few outstanding points:

- 1) 'Convenience samples' - please define this term in the text as it is not one that is universally used.
- 2) Please cite Fig. 1 in the text (preceding Fig. 2).
- 3) Page 3, para 3, line 11; change 'had' to 'have'
- 4) Para 4: Fig. 2 legend only refers to sequences from ticks

Reviewer #2 (Remarks to the Author):

None (Editor's note: takes issue with the nomenclature used in the current version)

Reviewer #3 (Remarks to the Author):

The revised ms addresses some of the prior concerns but some of the most important ones are left unaddressed.

The first of these is the inadequate revision of the potential for non-shrew hosts to act as reservoirs, given the detection of DTV in ticks having fed from them. Other non-shrew (and non-mouse) hosts have been shown to produce DTV-infected ticks, along with the present study, but the suggestion to address this was ignored. The authors' response to the original suggestion that they avoid suggesting a sole reservoir host for DTV is evasive. They argue that their detection of a single positive tick from "bird", "cat" and "Sciuridae" is inconclusive, given the potential for pre-feeding (i.e. transovarial) infection, but they fail to subject this assertion to a rigorous analysis. DTV infection prevalence in field-collected larvae (from their sites or from the published literature) can be used to determine the binomial probability that each host taxon could have produced the observed number of infected nymphal ticks as a result of those ticks having been previously infected as larvae. They could then identify reservoirs as those host taxa from which ticks were infected at a rate sufficiently high that it was unlikely (<5% chance) that the ticks had simply retained infection acquired transovarially.

The second is their refusal to use the correct nomenclature for the tick under study. Two of the three expert reviewers point out this easily-corrected error, but the authors refuse to make this simple revision. Their first reason they provide is that they can call the tick what they like if they disagree with the accepted systematics and nomenclature. Their second reason consists of their promise that the nomenclature will be changed in the near future based on unspecified accumulating evidence. Their third reason is that one of the many published criteria for abandoning the junior subjective synonym of "Ixodes dammini", namely the hybridization experiment of Oliver et al, "is not a good criterion for distinguishing between Ixodes species.", while failing to mention all the other criteria, which in aggregate led to the nomenclatural change. Their fourth reason is their claim that the use of multiple names for the same species, which they would like to perpetuate, is not confusing, providing no evidence for this, and asserting contra the Entomological Society of America and standard practice in systematics that "a common name is simply one that is commonly used." They oddly assert that "Deer tick is more commonly used than blacklegged tick by the public as well as scientists in the northeastern U.S.". Even if this assertion had support, it would be entirely irrelevant. Their fifth reason is perhaps the most surprising of all. They state, "We fail to see how the nomenclature influences the scientific merit of our findings." I cannot understand why the use of accepted scientific nomenclature, which has been a basic requirement in scientific publishing for well over a century, would NOT influence scientific merit. Moreover, if one were to accept their assertion that scientific nomenclature has no bearing on scientific merit, then why would they not simply make this correction, as requested by the majority of expert reviewers?

Reviewer #1 (Remarks to the Author):

Thank you for your constructive responses to the comments. I just have a few outstanding points:

1) 'Convenience samples' - please define this term in the text as it is not one that is universally used.

Line 63- We have removed the term from the sentence since it is obviously causing confusion. The new sentence, "Samples of host-seeking nymphal deer ticks were collected....." adequately conveys the point we are trying to make. Further details on the collections can be found in the methods section. (paragraph starting on 148)

2) Please cite Fig. 1 in the text (preceding Fig. 2).

This was added to the text on line 64.

3) Page 3, para 3, line 11; change 'had' to 'have'

Done

4) Para 4: Fig. 2 legend only refers to sequences from ticks

Thanks for noticing that. We have added "as well as the positive shrew..." to the legend to clarify that the sequence data from the shrew is included also.

Reviewer #2 (Remarks to the Author):

None (Editor's note: takes issue with the nomenclature used in the current version)

Reviewer #3 (Remarks to the Author):

The revised ms addresses some of the prior concerns but some of the most important ones are left unaddressed.

The first of these is the inadequate revision of the potential for non-shrew hosts to act as reservoirs, given the detection of DTV in ticks having fed from them. Other non-shrew (and non-mouse) hosts have been shown to produce DTV-infected ticks, along with the present study, but the suggestion to address this was ignored.

We regret that we have failed to mention the study by Dupuis et al in 2013 where infected larvae were removed from 2 skunks, 2 opossums, and 1 raccoon. We overlooked the finding of xenodiagnostic ticks from these hosts due to the paper's emphasis on the serologic results; the paper should have emphasized the infectivity of these hosts! We have added this critical reference to our manuscript. Line 139 now reads, "Virus has been detected from xenodiagnostic ticks removed from skunks, raccoons and opossum in New York (Dupuis et al. 2013). As with other tick-transmitted infections, contributions to the DTV enzootic cycle are likely to be dependent on local conditions and other hosts than shrews may contribute to maintenance. However, the association of shrews with DTV-infected ticks across multiple transmission seasons and across diverse sites, suggests that additional studies of shrews would be useful."

The authors' response to the original suggestion that they avoid suggesting a sole reservoir host for DTV is evasive. They argue that their detection of a single positive tick from "bird", "cat" and "Sciuridae" is inconclusive, given the potential for pre-feeding (i.e. transovarial) infection, but they fail to subject this assertion to a rigorous analysis. DTV infection prevalence in field-collected larvae (from their sites or from the published literature) can be used to determine the binomial probability that each host taxon could have produced the observed number of infected nymphal ticks as a result of those ticks having been previously infected as larvae. They could then identify reservoirs as those host taxa from which ticks were infected at a rate sufficiently high that it was unlikely (<5% chance) that the ticks had simply retained infection acquired transovarially.

The larvae that gave rise to the nymphs collected in our study would have been questing during late summer of 2017-2019. We have never found it useful to collect host-seeking larvae, so we regret that we cannot determine what the underlying rate of TOT was for the ticks included in our study. TOT has been experimentally demonstrated for POWV, but the vertical transmission rate (a product of the proportion of larvae infected within an egg batch and the proportion of ticks giving rise to at least one infected larva), which is more relevant to enzootic maintenance, has not been determined. - We anticipate that VTR will be no greater than for any other of the TBEV complex, estimated to be 0.1% (although it is not clear that this is VTR; Hartemink et al 2008). To determine whether the occurrence of shrews as a bloodmeal host is greater than what would have been expected to occur via TOT alone, we have added up the number of ticks used in this study (990 total ticks) and determined the 95% confidence interval around the expected prevalence of 0.1%, which would be 0 to 0.4%. This means that only 4 infected ticks are expected by TOT alone. We identified 13 ticks from shrews and thus, some other mode of transmission must also

contribute. Ticks derived from the other identified hosts fall within the probability of TOT transmission estimates. We added this to the manuscript.

Line 84 now reads, "Using the 0.1% estimated rate of transmission of adult female ticks to larval progeny for the related tick-borne encephalitis virus, we estimate that up to 4 ticks (95% binomial confidence interval of 0 to 0.4% of ticks) from our study could be derive from inheritance. More than 4 infected ticks were derived from shrews, suggesting that inheritance alone cannot explain the apparent association. The infected ticks derived from the other identified hosts fall within the probability of TOT transmission estimates."

Line 108 now reads, "We found that a greater number of ticks were associated with a specific host from all study sites than expected by vertical transmission, indicating that these ticks were not likely to have inherited the infection."

The second is their refusal to use the correct nomenclature for the tick under study. Two of the three expert reviewers point out this easily-corrected error, but the authors refuse to make this simple revision. Their first reason they provide is that they can call the tick what they like if they disagree with the accepted systematics and nomenclature. Their second reason consists of their promise that the nomenclature will be changed in the near future based on unspecified accumulating evidence. Their third reason is that one of the many published criteria for abandoning the junior subjective synonym of "Ixodes dammini", namely the hybridization experiment of Oliver et al, "is not a good criterion for distinguishing between Ixodes species.", while failing to mention all the other criteria, which in aggregate led to the nomenclatural change. Their fourth reason is their claim that the use of multiple names for the same species, which they would like to perpetuate, is not confusing, providing no evidence for this, and asserting contra the Entomological Society of America and standard practice in systematics that "a common name is simply one that is commonly used." They oddly assert that "Deer tick is more commonly used than blacklegged tick by the public as well as scientists in the northeastern U.S.". Even if this assertion had support, it would be entirely irrelevant. Their fifth reason is perhaps the most surprising of all. They state, "We fail to see how the nomenclature influences the scientific merit of our findings." I cannot understand why the use of accepted scientific nomenclature, which has been a basic requirement in scientific publishing for well over a century, would NOT influence scientific merit. Moreover, if one were to accept their assertion that scientific nomenclature has no bearing on scientific merit, then why would they not simply make this correction, as requested by the majority of expert reviewers?

We continue to maintain that the ICZN Code, the recognized global authority for animal nomenclature, allows for dissent in nomenclature. In the interests of publishing our study in a timely manner, we have changed the nomenclature as required. We note that science is often driven by questioning dogma and find it unusual that a reviewer would force colleagues to conform to the majority opinion; the senior author's >200 publications on deer tick-transmitted infections, in which the offending nomenclature has generally been used, is evidence that reviewers have ultimately recognized the value of scientific dissent even though they strongly disagreed. The merit of our manuscript is solely based on the likely contribution of shrews to the enzootic cycle of DTV and is based on a major advance in the field of "tick borne disease ecology", viz., the use of a novel host bloodmeal analysis tool. Nomenclature is not a focus of the study at all.